# End-to-end Symmetry Preserving Inter-atomic Potential Energy Model for Finite and Extended Systems

**Linfeng Zhang[1], Jiequn Han[1], Han Wang[2,3,*], Wissam A. Saidi[4,†],**
**Roberto Car[1,5,6], Weinan E[1,7,8,‡]**

[1] Program in Applied and Computational Mathematics, Princeton University, USA
[2] Institute of Applied Physics and Computational Mathematics, China
[3] CAEP Software Center for High Performance Numerical Simulation, China
[4] Department of Mechanical Engineering and Materials Science, University of Pittsburgh, USA
[5] Department of Chemistry and Department of Physics, Princeton University, USA
[6] Princeton Institute for the Science and Technology of Materials, Princeton University, USA
[7] Department of Mathematics, Princeton University, USA
[8] Beijing Institute of Big Data Research, China
*wang_han@iapcm.ac.cn, †alsaidi@pitt.edu, ‡weinan@math.princeton.edu

## Abstract

Machine learning models are changing the paradigm of molecular modeling, which is a fundamental tool for material science, chemistry, and computational biology. Of particular interest is the inter-atomic potential energy surface (PES). Here we develop Deep Potential - Smooth Edition (DeepPot-SE), an end-to-end machine learning-based PES model, which is able to efficiently represent the PES of a wide variety of systems with the accuracy of *ab initio* quantum mechanics models. By construction, DeepPot-SE is extensive and continuously differentiable, scales linearly with system size, and preserves all the natural symmetries of the system. Further, we show that DeepPot-SE describes finite and extended systems including organic molecules, metals, semiconductors, and insulators with high fidelity.

## 1 Introduction

Representing the inter-atomic potential energy surface (PES), both accurately and efficiently, is one of the most challenging problems in molecular modeling. Traditional approaches have either resorted to direct application of quantum mechanics models such as density functional theory (DFT) models [1, 2], or empirically constructed atomic potential models such as the embedded atomic method (EAM) [3]. The former approach is severely limited by the size of the system that one can handle, while as the latter class of methods are limited by the accuracy and the transferability of the model. This dilemma has confronted the molecular modeling community for several decades. In recent years, machine learning (ML) methods tackled this classical problem and a large body of work has been published in this area [4–17]. These studies have clearly demonstrated the potential of using ML methods and particularly neural network models to represent the PES. Considering the importance of the PES in molecular modeling, more work is needed to provide a general framework for an ML-based PES that can equally describe different systems with high fidelity.

Before proceeding further, let us list the requirements of the PES models that we consider to be fundamental: 1) The model should have the potential to be as accurate as quantum mechanics for both finite and extended systems. By finite system we mean that the system is isolated and surrounded by vacuum, e.g., gas-state molecules; by extended system we mean that the system is in a simulation cell

subject to periodic boundary conditions. 2) The only input for a PES model should be the chemical species and the atomic coordinates. Use of other input information should be avoided. 3) The PES model should be size extensive, i.e., if a system is composed of $A$ and $B$ subsystems, its energy should be close to the sum of $A$'s and $B$'s energies. This property is essential for handling different bulk systems with varying sizes. 4) The PES model should preserve the natural symmetries of the system, such as translational, rotational, and permutational symmetries. 5) Human intervention should be minimized. In other words, the model should be end-to-end. This is particularly relevant for multi-component or multi-phase systems, since we typically have limited knowledge about suitable empirical descriptors for these systems. 6) The model should be reasonably smooth, typically continuously differentiable such that forces are properly defined for molecular dynamics simulation.

In other words, from the viewpoint of a practitioner, the model should be comparable to first-principles quantum mechanical models in its ease-to-use and accuracy but at a significantly lesser computational cost.

Existing ML models generally satisfy only a subset of the above requirements. The Bonds-in-Molecules Neural Network method (BIM-NN) [15], for example, uses empirical information on the chemical bonds as input, violating requirement 2). The Gradient Domain Machine Learning (GDML) scheme [11] uses a global descriptor for the whole molecular pattern, violating 3). The Deep Potential model [16, 17] represents the PES as a sum of "atomic" energies that depend on the coordinates of the atoms in each atomic environment in a symmetry-preserving way. This is achieved, however, at the price of introducing discontinuities in the model, thus violating 6). The Behler-Parrinello Neural Network (BPNN) model [4] uses hand-crafted local symmetry functions as descriptors. These require human intervention, violating 5).

From the viewpoint of supervised learning, there have been many interesting and challenging large-scale examples for classification tasks, but relatively few for regression. In this regard, the PES provides a natural candidate for a challenging regression task.

The main contributions of this paper are twofolds. First, we propose and test a new PES model that satisfies all the requirements listed above. We call this model Deep Potential – Smooth Edition (DeepPot-SE). We believe that the methodology proposed here is also applicable to other ML tasks that require a symmetry-preserving procedure. Second, we test the DeepPot-SE model on various systems, which extend previous studies by incorporating DFT data for challenging materials such as high entropy alloys (HEAs). We used the DeePMD-kit package [18] for all training and testing tasks. The corresponding code[1] and data[2] are released online.

## 2    Related Work

*Spherical CNN and DeepSets.* From the viewpoint of preserving symmetries, the Spherical CNN [19] and DeepSets [20] models are the most relevant to our work. The spherical CNN model incorporates the definition of $S^2$ and $SO(3)$ cross-correlations and has shown impressive performance in preserving rotational invariance. The DeepSets model provides a family of functions to which any permutation invariant objective function must belong and has been tested on several different tasks, including population statistic estimation, point cloud classification, set expansion, and outlier detection.

*ML-based PES models.* In addition to the previously mentioned BIM-NN, BPNN, DeepPot, and GDML approaches, some other ML models for representing the PES include: The Smooth Overlap of Atomic Positions model (SOAP) [21] uses a kernel method based on a smooth similarity measure of two neighboring densities. The Deep Tensor Neural Network (DTNN) model [10] uses as input a vector of nuclear charges and an inter-atomic distance matrix, and introduces a sequence of interaction passes where "the atom representations influence each other in a pair-wise fashion". Recently, the SchNet model [12] proposed a new continuous-filter convolutional layer to model the local atomic correlations and successfully modeled quantum interactions in small molecules.

## 3 Theory

### 3.1 Preliminaries

Consider a system of $N$ atoms, $\boldsymbol{r} = \{\boldsymbol{r}_1, \boldsymbol{r}_2, ..., \boldsymbol{r}_N\}$, in a 3-dimensional Euclidean space. We define the coordinate matrix $\mathcal{R} \in \mathbb{R}^{N \times 3}$, whose $i$th column contains the 3 Cartesian coordinates of $\boldsymbol{r}_i$, i.e.,

$$\mathcal{R} = \{\boldsymbol{r}_1^T, \cdots, \boldsymbol{r}_i^T, \cdots, \boldsymbol{r}_N^T\}^T, \; \boldsymbol{r}_i = (x_i, y_i, z_i). \tag{1}$$

The PES $E(\mathcal{R}) \equiv E$ is a function that maps the atomic coordinates and their chemical characters to a real number. Using the energy function $E$, we define the force matrix $\mathcal{F}(\mathcal{R}) \equiv \mathcal{F} \in \mathbb{R}^{N \times 3}$ and the $3 \times 3$ virial tensor $\Xi(\mathcal{R}) \equiv \Xi$ by:

$$\mathcal{F} = -\nabla_\mathcal{R} E \; \left(\mathcal{F}_{ij} = -\nabla_{\mathcal{R}_{ij}} E\right), \text{ and } \Xi = \text{tr}[\mathcal{R} \otimes \mathcal{F}] \; \left(\Xi_{ij} = \sum_{k=1}^{N} \mathcal{R}_{ki} \mathcal{F}_{kj}\right), \tag{2}$$

respectively. Finally, we denote the full parameter set used to parametrize $E$ by $\boldsymbol{w}$, and we write the corresponding PES model as $E^{\boldsymbol{w}}(\mathcal{R}) \equiv E^{\boldsymbol{w}}$. The force $\mathcal{F}^{\boldsymbol{w}}$ and the virial $\Xi^{\boldsymbol{w}}$ can be directly computed from $E^{\boldsymbol{w}}$.

As illustrated in Fig. 1, in the DeepPot-SE model, the extensive property of the total energy is preserved by decomposing it into "atomic contributions" that are represented by the so-called sub-networks, i.e.:

$$E^{\boldsymbol{w}}(\mathcal{R}) = \sum_i E^{\boldsymbol{w}_{\alpha_i}}(\mathcal{R}^i) \equiv \sum_i E_i, \tag{3}$$

where $\alpha_i$ denotes the chemical species of atom $i$. We use the subscript $(...)^{\boldsymbol{w}_{\alpha_i}}$ to show that the parameters used to represent the "atomic energy" $E_i$ depend only on the chemical species $\alpha_i$ of atom $i$. Let $r_c$ be a pre-defined cut-off radius. For each atom $i$, we consider its neighbors $\{j | j \in \mathcal{N}_{r_c}(i)\}$, where $\mathcal{N}_{r_c}(i)$ denotes the atom indices $j$ such that $r_{ji} < r_c$, with $r_{ji}$ being the Euclidean distance between atoms $i$ and $j$. We define $N_i = |\mathcal{N}_{r_c}(i)|$, the cardinality of the set $\mathcal{N}_{r_c}(i)$, and use $\mathcal{R}^i \in \mathbb{R}^{N_i \times 3}$ to denote the local environment of atom $i$ in terms of Cartesian coordinates:

$$\mathcal{R}^i = \{\boldsymbol{r}_{1i}^T, \cdots, \boldsymbol{r}_{ji}^T, \cdots, \boldsymbol{r}_{N_i,i}^T\}^T, \; \boldsymbol{r}_{ji} = (x_{ji}, y_{ji}, z_{ji}). \tag{4}$$

Note that here $\boldsymbol{r}_{ji} \equiv \boldsymbol{r}_j - \boldsymbol{r}_i$ are defined as *relative* coordinates and the index $j$ $(1 \le j \le N_i)$ is used to denote the neighbors of the $i$th atom. Correspondingly, we have $r_{ji} = \|\boldsymbol{r}_{ji}\|$.

The construction in Eq. (3) is shared by other empirical potential models such as the EAM method [3], and by many size-extensive ML models like the BPNN method [4]. However, these approaches differ in the representation of $E_i$.

The sub-network for $E_i$ consists of an encoding and a fitting neural network. The encoding network is specially designed to map the local environment $\mathcal{R}^i$ to an embedded feature space, which preserves the translational, rotational, and permutational symmetries of the system. The fitting network is a fairly standard fully-connected feedforward neural network with skip connections, which maps the embedded features to an "atomic energy". The optimal parameters for both the encoding and fitting networks are obtained by a single end-to-end training process to be specified later.

### 3.2 Construction of symmetry preserving functions

Before going into the details of the sub-network for $E_i$, we consider how to represent a scalar function $f(\boldsymbol{r})$, which is invariant under translation, rotation, and permutation, i.e.:

$$\hat{T}_{\boldsymbol{b}} f(\boldsymbol{r}) = f(\boldsymbol{r} + \boldsymbol{b}), \; \hat{R}_\mathcal{U} f(\boldsymbol{r}) = f(\boldsymbol{r}\mathcal{U}), \; \hat{P}_\sigma f(\boldsymbol{r}) = f(\boldsymbol{r}_{\sigma(1)}, \boldsymbol{r}_{\sigma(2)}, ..., \boldsymbol{r}_{\sigma(N)}), \tag{5}$$

respectively. Here $\boldsymbol{b} \in \mathbb{R}^3$ is an arbitrary 3-dimensional translation vector, $\mathcal{U} \in \mathbb{R}^{3 \times 3}$ is an orthogonal rotation matrix, and $\sigma$ denotes an arbitrary permutation of the set of indices.

Granted the fitting ability of neural networks, the key to a general representation is an *embedding* procedure that maps the original input $\boldsymbol{r}$ to symmetry preserving components. The embedding components should be faithful in the sense that their pre-image should be equal to $\boldsymbol{r}$ up to a symmetry operation. We draw inspiration from the following two observations.

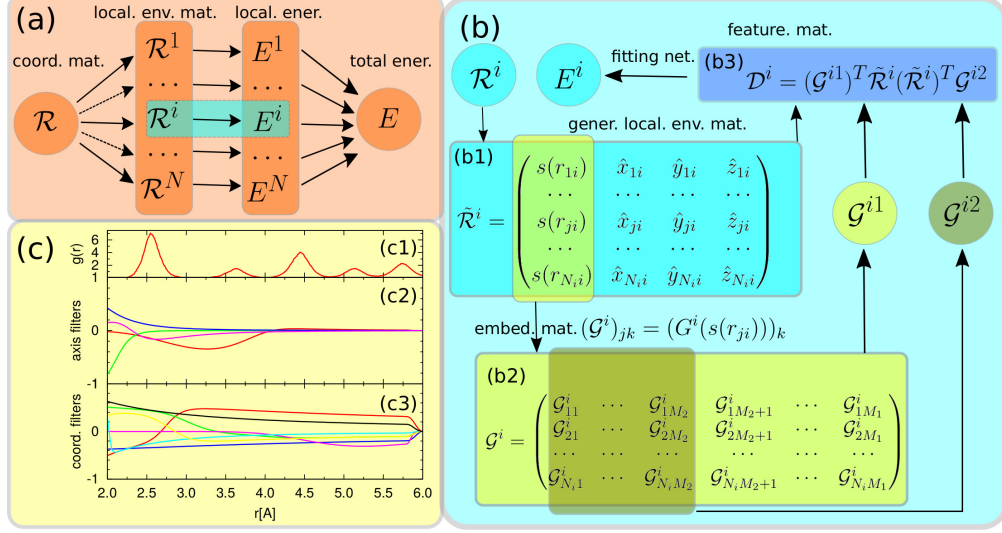

Figure 1: Schematic plot of the DeepPot-SE model. (a) The mapping from the coordinate matrix $\mathcal{R}$ to the PES $E$. First, $\mathcal{R}$ is transformed to local environment matrices $\{\mathcal{R}^i\}_{i=1}^N$. Then each $\mathcal{R}^i$ is mapped, through a sub-network, to a local "atomic" energy $E_i$. Finally, $E = \sum_i E_i$. (b) The zoom-in of a sub-network. (b1) The transformation from $\mathcal{R}^i$ to the generalized local environment matrix $\tilde{\mathcal{R}}^i$; (b2) The radial part of $\tilde{\mathcal{R}}^i$ is mapped, through an encoding network, to the embedding matrix $\mathcal{G}^{i1} \in \mathbb{R}^{N_i \times M_1}$ and $\mathcal{G}^{i2} \in \mathbb{R}^{N_i \times M_2}$; (b3) The $M_1 \times M_2$ symmetry preserving features, contained in $\mathcal{D}^i$, are given by the matrix product of $(\mathcal{G}^{i1})^T$, $\tilde{\mathcal{R}}^i$, $(\tilde{\mathcal{R}}^i)^T$, and $\mathcal{G}^{i2}$. (c) Illustrative plot of the embedding function $\mathcal{G}^i$, taking Cu as an example. (c1) radial distribution function $g$ of the training data; (c2) $M_2$ $(=4)$ axis filters, defined as the product of $\mathcal{G}^{i2}$ and $s(r)$, as functions of $r$; (c3) 6 out of $M_1$ $(=80)$ coordinate filters, defined as the product of $\mathcal{G}^{i1}$ and $s(r)$, as functions of $r$.

*Translation and Rotation.* For each object $i$, the symmetric matrix

$$\Omega^i \equiv \mathcal{R}^i(\mathcal{R}^i)^T \qquad (6)$$

is an over-complete array of invariants with respect to translation and rotation [21, 22], i.e., it contains the complete information of the neighboring point pattern of atom $i$. However, this symmetric matrix switches rows and columns under a permutational operation.

*Permutation.* Theorem 2 of Ref. [20] states that any permutation symmetric function $f(\boldsymbol{r})$ can be represented in the form $\rho(\sum_i \phi(\boldsymbol{r}_i))$, where $\phi(\boldsymbol{r}_i)$ is a multidimensional function, and $\rho(...)$ is another general function. For example,

$$\sum_i g(\boldsymbol{r}_i)\boldsymbol{r}_i \qquad (7)$$

is invariant under permutation for any scalar function $g$.

### 3.3 The DeepPot-SE sub-networks

As shown in Fig. 1, we construct the sub-networks in three steps. First, the relative coordinates $\mathcal{R}^i \in \mathbb{R}^{N_i \times 3}$ are mapped onto generalized coordinates $\tilde{\mathcal{R}}^i \in \mathbb{R}^{N_i \times 4}$. In this mapping, each row of $\mathcal{R}^i$, $\{x_{ji}, y_{ji}, z_{ji}\}$, is transformed into a row of $\tilde{\mathcal{R}}^i$:

$$\{x_{ji}, y_{ji}, z_{ji}\} \mapsto \{s(r_{ji}), \hat{x}_{ji}, \hat{y}_{ji}, \hat{z}_{ji}\}, \qquad (8)$$

where $\hat{x}_{ji} = \frac{s(r_{ji})x_{ji}}{r_{ji}}$, $\hat{y}_{ji} = \frac{s(r_{ji})y_{ji}}{r_{ji}}$, $\hat{z}_{ji} = \frac{s(r_{ji})z_{ji}}{r_{ji}}$, and $s(r_{ji}) : \mathbb{R} \mapsto \mathbb{R}$ is a continuous and differentiable scalar weighting function applied to each component, defined as:

$$s(r_{ji}) = \begin{cases} \dfrac{1}{r_{ji}}, & r_{ji} < r_{cs}. \\ \dfrac{1}{r_{ji}}\left\{\dfrac{1}{2}\cos\left[\pi\dfrac{(r_{ji} - r_{cs})}{(r_c - r_{cs})}\right] + \dfrac{1}{2}\right\}, & r_{cs} < r_{ji} < r_c. \\ 0, & r_{ji} > r_c. \end{cases} \qquad (9)$$

Here $r_{cs}$ is a smooth cutoff parameter that allows the components in $\tilde{\mathcal{R}}^i$ to smoothly go to zero at the boundary of the local region defined by $r_c$. The weighting function $s(r_{ji})$ reduces the weight of the particles that are more distant from atom $i$. In addition, it removes from the DeepPot-SE model the discontinuity introduced by the cut-off radius $r_c$.

Next, we define the local embedding network $G^{\alpha_j, \alpha_i}(s(r_{ji}))$, shorthanded as $G(s(r_{ji}))$, a neural network mapping from a single value $s(r_{ji})$, through multiple hidden layers, to $M_1$ outputs. Note that the network parameters of $G$ depend on the chemical species of both atom $i$ and its neighbor atom $j$. The local embedding matrix $\mathcal{G}^i \in \mathbb{R}^{N_i \times M_1}$ is the matrix form of $G(s(r_{ji}))$:

$$(\mathcal{G}^i)_{jk} = (G(s(r_{ji})))_k. \qquad (10)$$

Observe that $\tilde{\mathcal{R}}^i(\tilde{\mathcal{R}}^i)^T$ is a generalization of the symmetry matrix $\Omega^i$ in Eq. (6) that preserves rotational symmetry, and $(\mathcal{G}^i)^T\tilde{\mathcal{R}}^i$ is a special realization of the permutation invariant operations in Eq. (7). This motivates us to define, finally, the encoded feature matrix $\mathcal{D}^i \in \mathbb{R}^{M_1 \times M_2}$ of atom $i$:

$$\mathcal{D}^i = (\mathcal{G}^{i1})^T \tilde{\mathcal{R}}^i (\tilde{\mathcal{R}}^i)^T \mathcal{G}^{i2} \qquad (11)$$

that preserves both the rotation and permutation symmetry. Here $\mathcal{G}^{i1}$ and $\mathcal{G}^{i2}$ are matrices of the form (10). Apparently the translation symmetry is meanwhile preserved in (11).

In practice, we take $\mathcal{G}^{i1} = \mathcal{G}^i$ and take the first $M_2$ ($< M_1$) columns of $\mathcal{G}^i$ to form $\mathcal{G}^{i2} \in \mathbb{R}^{N_i \times M_2}$. Lastly, the $M_1 \times M_2$ components contained in the feature matrix $\mathcal{D}^i$ are reshaped into a vector to serve as the input of the fitting network, and yield the "atomic energy" $E_i$. In the Supplementary Materials, we show explicitly that $\mathcal{D}^i$, and hence the DeepPot-SE model, preserves all the necessary symmetries. Moreover, DeepPot-SE model has a linear scaling with respect to $N$ in computational complexity. Suppose there are at most $N_c$ neighboring atoms within the cut-off radius of each atom and the complexity in evaluating the atomic energy $E_i$ is $f(N_c)$, then according to the local energy decomposition of PES, the total complexity of the model is $\sim f(N_c)N$. No matter how large $N$ is, $N_c$ only depends on $R_c$ and is essentially bounded due to physical constraints.

We remark that, considering the explanation of the Deep Potential [16] and the fact that $M_1$ is much larger than $M_2$ in practice, we view the role of $(\mathcal{G}^{i1})^T\tilde{\mathcal{R}}^i$ as being the mapping from the atomic point pattern to a feature space that preserves permutation symmetry. The role of $(\tilde{\mathcal{R}}^i)^T\mathcal{G}^{i2}$ is to select symmetry-preserving axes onto which $(\mathcal{G}^{i1})^T\tilde{\mathcal{R}}^i$ is projected . Therefore, we call $\mathcal{G}^{i1}$ the coordinate filters and $\mathcal{G}^{i2}$ the axis filters. More specifically, each output of the embedding network $\mathcal{G}^i$ can be thought of as a distance-and chemical-species-dependent filter, which adds a weight to the neighboring atoms. To provide an intuitive idea of $\mathcal{G}^1$ and $\mathcal{G}^2$, we show in Fig. 1(c) the results of these filters obtained after training to model crystalline Cu at finite temperatures. To help understand these results, we also display the radial distribution function, $g(r)$, of Cu. It is noted that unlike the fixed filters such as Gaussians, these embedded filters are adaptive in nature. Generally, we have seen that choosing $M_1 \sim 100$, which is of the order of the number of neighbors of each atom within the cutoff radius $r_c$, and $M_2 \sim 4$, gives good empirical performance. As shown by Fig. 1(c), for Cu, the $M_2 = 4$ outputs of $\mathcal{G}^{i2}$ mainly give weights to neighbors within the first two shells, i.e., the first two peaks of $g(r)$, while the shapes of other filters, as outputs of $\mathcal{G}^{i1}$, are more diversified and general.

### 3.4 The training process

The parameters $\boldsymbol{w}$ contained in the encoding and fitting networks are obtained by a training process with the Adam stochastic gradient descent method [23]. We define a family of loss functions,

$$L(p_\epsilon, p_f, p_\xi) = \frac{1}{|\mathcal{B}|}\sum_{l \in \mathcal{B}} p_\epsilon |E_l - E_l^{\boldsymbol{w}}|^2 + p_f|\mathcal{F}_l - \mathcal{F}_l^{\boldsymbol{w}}|^2 + p_\xi ||\Xi_l - \Xi_l^{\boldsymbol{w}}||^2. \qquad (12)$$

Here $\mathcal{B}$ denotes the minibatch, $|\mathcal{B}|$ is the batch size, $l$ denotes the index of the training data, which typically consists of the snapshot of the atomic configuration (given by the atomic coordinates, the atomic species, and the cell tensor), and the labels (the energy, the force, and the virial). In Eq. (12), $p_\epsilon$, $p_f$, and $p_\xi$ are tunable prefactors. When one or two labels are missing from the data, we set the corresponding prefactor(s) to zero. It is noted that the training process is trying to maximize the usage of the training data. Using only the energy for training should, in principle, gives a good PES model. However, the use of forces in the training process significantly reduces the number of snapshots needed to train a good model.

## 4   Data and Experiments

We test the DeepPot-SE model on a wide variety of systems comprising molecular and extended systems. The extended systems include single- and multi-element metallic, semi-conducting, and insulating materials. We also include supported nanoparticles and HEAs, which constitute very challenging systems to model. See Table 2 for a general view of the data. The data of molecular systems are from Refs. [10, 11] and are available online [3]. The data of $C_5H_5N$ (pyridine) are from Ref. [24]. We generated the rest of the data using the CP2K package [25]. For each system, we used a large super cell constructed from the optimized unit cell. The atomic structures are collected from different *ab initio* molecular trajectories obtained from NVT ensemble simulations with temperature ranging from 100 to 2000 K. To minimize correlations between the atomic configurations in the *ab initio* MD trajectories, we swapped atomistic configurations between different temperatures or randomly displaced the atomic positions after 1 ps. Furthermore, to enhance the sampling of the configuration space, we used a relatively large time step of 10 fs, even though this increased the number of steps to achieve self-consistency for solving the Kohn-Sham equations [1] at each step. More details of each extended system are introduced in Section 4.2 and the corresponding data description is available online in the data reservoir[4].

For clarification, we use the term *system* to denote a set of data on which a unified DeepPot-SE model is fitted, and use the term *sub-system* to denote data with different composition of atoms or different phases within a system. For all systems, we also test the DeePMD model for comparison, which is more accurate and robust than the original Deep Potential model [16]. The network structure and the training scheme (learning rate, decay step, etc.) are summarized in the Supplementary Materials.

### 4.1   Small organic molecules

| molecule | DeepPot-SE | DeePMD [17] | GDML [11] | SchNet [12] |
|---|---|---|---|---|
| Aspirin | 6.7, **12.1** (10.2, 19.4) | 8.7, 19.1 | 11.7, 42.9 | **5.2**, 14.3 |
| Ethanol | **2.2**, 3.1 (3.1, 7.7) | 2.4, 8.3 | 6.5, 34.3 | **2.2**, **2.2** |
| Malonaldehyde | **3.3**, 4.4 (4.7, 9.7) | 4.0, 12.7 | 6.9, 34.7 | 3.5, **3.5** |
| Naphthalene | 5.2, 5.5 (6.5, 13.1) | **4.1**, 7.1 | 5.2, 10.0 | 4.8, **4.8** |
| Salicylic acid | 5.0, **6.6** (6.3, 13.0) | 4.6, 10.9 | 5.2, 12.1 | 4.3, **8.2** |
| Toluene | 4.4, 5.8 (7.8, 13.3) | **3.7**, 8.5 | 5.2, 18.6 | 3.9, **3.9** |
| Uracil | 4.7, **2.8** (5.0, 9.2) | **3.7**, 9.8 | 4.8, 10.4 | 4.3, 4.8 |

Table 1: Mean absolute errors (MAEs) for energy and force predictions in meV and meV/Å, respectively, denoted by a pair of numbers in the table. Results obtained by the DeepPot-SE, DeePMD, GDML, and SchNet methods are summarized. Using the DeepPot-SE method, we trained both a unified model (results in brackets) that describes the seven molecular systems, and individual models that treat each molecule alone. The GDML and SchNet benchmarks are from Ref. [12]. SchNet, DeepPot-SE and DeePMD used 50,000 structures for training obtained from a molecular dynamics trajectory of small organic molecules. As explained in Ref. [12], GDML does not scale well with the number of atoms and training structures, and therefore used only 1000 structures for training. Best results among the considered models for each molecule are displayed in bold.

The small molecular system consists of seven different sub-systems, namely aspirin, ethanol, malonaldehyde, naphthalene, sallcylic acid, toluene, and uracil. The dataset has been benchmarked by

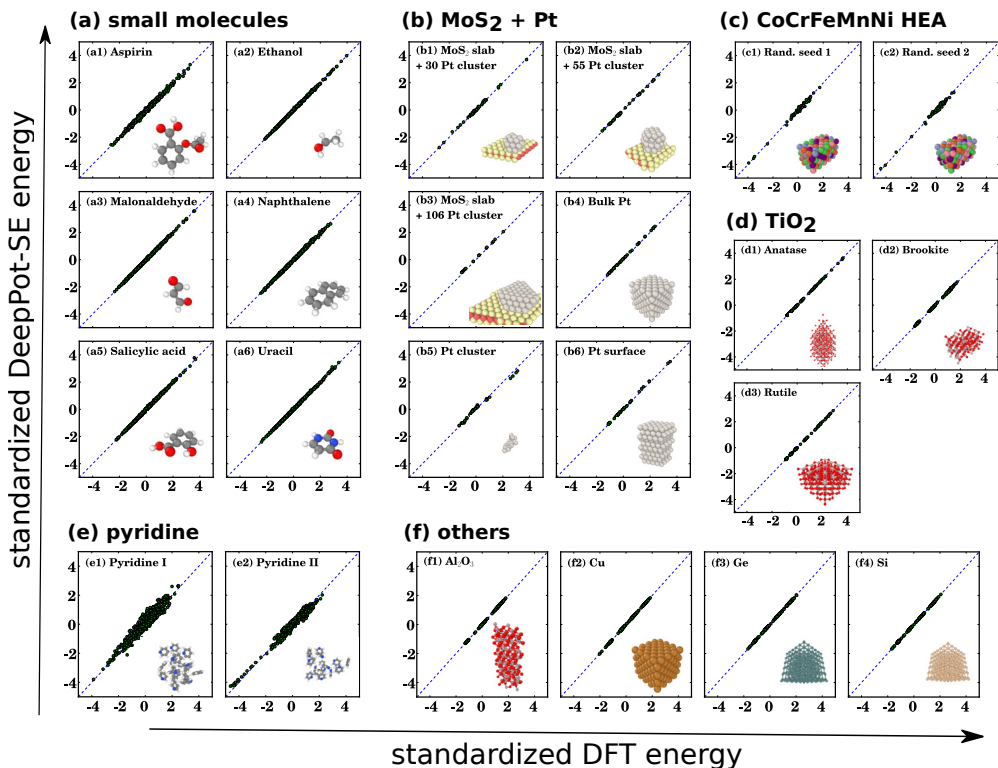

Figure 2: Comparison of the DFT energies and the DeepPot-SE predicted energies on the testing snapshots. The range of DFT energies of different systems is large. Therefore, for illustrative purpose, for each sub-system, we calculate the average $\mu_E$ and standard deviation $\sigma_E$ of DFT energies, and standardize both the DFT energies and the DeepPot-SE predicted energies by subtracting $\mu_E$ from them and then dividing them by $\sigma_E$. Then we plot the standardized energies within $\pm 4.5\sigma_E$. (a) The unified DeepPot-SE model for the small molecular system. These molecules contain up to 4 types of atoms, namely C, H, O, and N. Therefore, essentially 4 atomic sub-networks are learned and the corresponding parameters are shared by different molecules. (b) The DeepPot-SE model for the $MoS_2$ and Pt system. To make it robust for a real problem of structural optimization for Pt clusters on $MoS_2$ slabs, this model learn different sub-systems, in particular Pt clusters of various sizes on $MoS_2$ slabs. 6 representative sub-systems are selected in this figure. (c) The DeepPot-SE model for the CoCrFeMnNi HEA system. The sub-systems are different in random occupations of the elements on the lattice sites. 2 out of 48 sub-systems are selected in this figure. (d) The DeepPot-SE model for the $TiO_2$ system, which contains 3 different polymorphs. (e) The DeepPot-SE model for the pyridine ($C_5H_5N$) system, which contains 2 different polymorphs. (f) Other systems: $Al_2O_3$, Cu, Ge, and Si.

GDML, SchNet, and DeePMD [11, 12, 17]. Unlike previous models, our emphasis here is to train one *unified model* for all such molecules. A unified model can be used to study chemical reactions and could be transferable to unknown molecules. Therefore, it would be interesting and highly desirable to train a unified model for all of these sub-systems. The molecules in the dataset contain at most 4 different types of atoms, namely C, H, O, and N. Therefore, we need 4 sub-networks corresponding to the four types of atoms with different environments. We also compare the results of the unified model with the model trained individually for each sub-system. As shown in Table 1, all the methods show good performance in fitting both energies and forces of the small organic molecules. The MAEs of the total energy are in all cases below chemical accuracy (0.04 eV), a commonly used benchmark. The performance of the unified model is slightly worse than the individual models, but is still generally comparable.

| System | sub-system | # snapshot | Energy [meV] | Force [meV/Å] |
|---|---|---|---|---|
| bulk Cu | FCC solid | 3250 | **0.18** (0.25) | **90** (**90**) |
| bulk Ge | diamond solid | 4468 | **0.35** (0.60) | 38 (**35**) |
| bulk Si | diamond solid | 6027 | **0.24** (0.51) | 36 (**31**) |
| bulk $Al_2O_3$ | Trigonal solid | 5624 | **0.23** (0.48) | **49** (55) |
| bulk $C_5H_5N$ | Pyridine-I | 20121 | 0.38 (**0.25**) | **25** (**25**) |
| | Pyridine-II | 18103 | 0.65 (**0.43**) | **39** (**39**) |
| bulk $TiO_2$ | Rutile | 2779 | **0.96** (1.97) | **137** (163) |
| | Anatase | 2371 | **1.78** (3.37) | **181** (216) |
| | Brookite | 4877 | **0.59** (1.97) | **94** (109) |
| $MoS_2$+Pt | $MoS_2$ slab | 555 | **5.26** (17.2) | **23** (34) |
| | bulk Pt | 1717 | 2.00 (**1.85**) | **84** (226) |
| | Pt surface | 2468 | **6.77** (7.12) | **105** (187) |
| | Pt cluster | 927 | **30.6** (35.4) | **201** (255) |
| | Pt on $MoS_2$[a] | 46915 | **2.62** (5.89) | **94** (127) |
| CoCrFeMnNi HEA | rand. occ. I[b] | 13910 | **1.68** (6.99) | **394** (481) |
| | rand. occ. II[c] | 958 | **5.29** (21.7) | **410** (576) |

[a]Since Pt clusters have different sizes, this case contains more than one sub-system. The reported values are averages of all the sub-systems.

[b]This case includes 40 different random occupations of the elements on the lattice sites of the HEA system within the training dataset.

[c]This case includes 16 other random occupations that are different from the training dataset.

Table 2: The number of snapshots and the root mean square error (RMSE) of the DeepPot-SE prediction for various systems in terms of energy and forces. The RMSEs of the energies are normalized by the number of atoms in the system. The numbers in parentheses are the DeePMD results. For all sub-systems, 90% randomly selected snapshots are used for training, and the remaining 10% are used for testing. Moreover, for the HEA system, more data corresponding to 16 random occupations that are significantly different from the training dataset are added into the test dataset. Better results are in bold.

## 4.2 Bulk systems

Bulk systems are more challenging ML tasks due to their extensive character. In addition, in many cases, difficulties also come from the complexity of the system under consideration. For example, for systems containing many different phases or many different atomic components, physical/chemical intuition can hardly be ascertained. This is an essential obstacle for constructing hand-crafted features or kernels. Here we prepare two types of systems for the dataset and present results obtained from both DeepPot-SE and DeePMD methods. The first type of systems includes Cu, Ge, Si, $Al_2O_3$, $C_5H_5N$, and $TiO_2$. These datasets serve as moderately challenging tasks for a general end-to-end method. For the second type of systems, we include supported $(Pt)_n$ ($n \leq 155$) nano-clusters on $MoS_2$ and a high entropy 5-element alloy. These are more challenging systems due to the different components of the atoms in the system. See Fig. 2 for illustration.

*General systems.* As shown in Table 2, the first type of systems Cu, Ge, Si, and $Al_2O_3$ only contain one single solid phase and are relatively easy. For these systems both the DeePMD and the DeeMD-SE methods yield good results. The cases of $C_5H_5N$ (pyridine) and $TiO_2$ are more challenging. There are two polymorphs, or phases, of crystalline $C_5H_5N$ called pyridine-I and pyridine-II, respectively (See their structures in Ref. [24]). There are three phases of $TiO_2$, namely rutile, anatase, and brookite. Both rutile and anatase have a tetragonal unit cell, while brookite has an orthorhombic unit cell.

*Grand-canonical-like system: Supported Pt clusters on a $MoS_2$ slab.* Supported noble metal nanometer clusters (NCs) play a pivotal role in different technologies such as nano-electronics, energy storage/conversion, and catalysis. Here we investigate supported Pt clusters on a $MoS_2$ substrate, which have been the subject of intense investigations recently [26–31]. The sub-systems include pristine $MoS_2$ substrate, bulk Pt, Pt (100), (110) and (111) surfaces, Pt clusters, and supported Pt clusters on a $MoS_2$ substrate. The size of the supported Pt clusters ranges from 6 to 20, and 30, 55, 82, 92, 106, 134, and 155 atoms. The multi-component nature of this system, the extended character of the substrate, and the different sizes of the supported clusters with grand-canonical-like features,

make this system very challenging for an end-to-end framework. Yet as shown in Table 2 and Fig. 2, a unified DeepPot-SE model is able to capture these effects with satisfactory accuracy.

*The CoCrFeMnNi HEA system.* HEA is a new class of emerging advanced materials with novel alloy design concept. In the HEA, five or more equi-molar or near equi-molar alloying elements are deliberately incorporated into a single lattice with random site occupancy [32, 33]. Given the extremely large number of potential configurations of the alloy, entropic contributions to the thermodynamic landscape dictate the stability of the system in place of the cohesive energy. The HEA poses a significant challenge for *ab initio* calculations due to the chemical disorder and the large number of spatial configurations. Here we focus on a CoCrFeMnNi HEA assuming equi-molar alloying element distribution. We employ a 3x3x5 supercell based on the FCC unit cell with different random distributions of the elements at the lattice sites. In our calculations we used the experimental lattice constant reported in Ref. [34]. Traditionally it has been hard to obtain a PES model even for alloy systems containing less than 3 components. As shown by Table 2, the DeepPot-SE model not only is able to fit snapshots with random allocations of atoms in the training data, but also show great promise in transferring to systems with random locations that seem significantly different from the training data.

# 5   Summary

In this paper, we developed DeepPot-SE, an end-to-end, scalable, symmetry preserving, and accurate potential energy model. We tested this model on a wide variety of systems, both molecular and periodic. For extended periodic systems, we show that this model can describe cases with diverse electronic structure such as metals, insulators, and semiconductors, as well as diverse degrees of complexity such as bulk crystals, surfaces, and high entropy alloys. In the future, it will be of interest to expand the datasets for more challenging scientific and engineering studies, and to seek strategies for easing the task of collecting training data. In addition, an idea similar to the feature matrix has been recently employed to solve many-electron Schrödinger equation [35]. It will be of interest to see the application of similar ideas to other ML-related tasks for which invariance under translation, rotation, and/or permutation plays a central role.

**Acknowledgments**

We thank the anonymous reviewers for their careful reading of our manuscript and insightful comments and suggestions. The work of L. Z., J. H., and W. E is supported in part by ONR grant N00014-13-1-0338, DOE grants DE-SC0008626 and DE-SC0009248, and NSFC grants U1430237 and 91530322. The work of R. C. is supported in part by DOE grant DE-SC0008626. The work of H. W. is supported by the National Science Foundation of China under Grants 11501039 and 91530322, the National Key Research and Development Program of China under Grants 2016YFB0201200 and 2016YFB0201203, and the Science Challenge Project No. JCKY2016212A502. W.A.S. acknowledges financial support from National Science Foundation (DMR-1809085). We are grateful for computing time provided in part by the Extreme Science and Engineering Discovery Environment (XSEDE), which is supported by National Science Foundation (# NSF OCI-1053575), the Argonne Leadership Computing Facility, which is a DOE Office of Science User Facility supported under Contract DE-AC02-06CH11357, the National Energy Research Scientific Computing Center (NERSC), which is supported by the Office of Science of the U.S. Department of Energy under Contract No. DE-AC02-05CH11231, and the Terascale Infrastructure for Groundbreaking Research in Science and Engineering (TIGRESS) High Performance Computing Center and Visualization Laboratory at Princeton University.

## Footnotes

[1]`https://github.com/deepmodeling/deepmd-kit`

[2]`http://www.deepmd.org/database/deeppot-se-data/`

[3]See http://www.quantum-machine.org

[4]http://www.deepmd.org/database/deeppot-se-data/

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
