[Supplementary Material · Supplementary_Materials_for___it_End_to_end_Symmetry_Preserving_Inter_atomic_Potential_Energy_Model_for_Finite_and_Extended_Systems_.pdf]

# Supplementary Materials for *End-to-end Symmetry Preserving Inter-atomic Potential Energy Model for Finite and Extended Systems*

**Linfeng Zhang**[1], **Jiequn Han**[1], **Han Wang**[2,3,*], **Wissam A. Saidi**[4,†],
**Roberto Car**[1,5,6], **Weinan E**[1,7,8,‡]

[1] Program in Applied and Computational Mathematics, Princeton University, USA
[2] Institute of Applied Physics and Computational Mathematics, China
[3] CAEP Software Center for High Performance Numerical Simulation, China
[4] Department of Mechanical Engineering and Materials Science, University of Pittsburgh, USA
[5] Department of Chemistry and Department of Physics, Princeton University, USA
[6] Princeton Institute for the Science and Technology of Materials, Princeton University, USA
[7] Department of Mathematics, Princeton University, USA
[8] Beijing Institute of Big Data Research, China
*wang_han@iapcm.ac.cn, †alsaidi@pitt.edu, ‡weinan@math.princeton.edu

Table S1: Details of network structure and training scheme for each system.

| System | enbedding net[a] | $M_2$[b] | fitting net | cutoff[c] | LR scheme[d] | pref. scheme[e] |
|---|---|---|---|---|---|---|
| Mol. uni.[f] | (50,100) | 6 | (240,240,240) | - | (2e+7,1e+5,32,5e-4,0.97) | (0.02,1.0,1000,1) |
| Mol. ind.[g] | (30,60) | 6 | (80,80,80) | - | (2e+7,1e+5,32,5e-3,0.96) | (0.2,1.0,1000,1) |
| Cu | (40,80) | 4 | (160,160,160) | (5.8,6.0) | (2e+6,1e+4,1,5e-3,0.95) | (0.1,0.1,1000,1) |
| Ge | (40,80) | 4 | (160,160,160) | (6.8,7.0) | (2e+6,1e+4,1,5e-3,0.95) | (0.1,0.1,1000,1) |
| Si | (40,80) | 4 | (160,160,160) | (5.8,6.0) | (2e+6,1e+4,1,5e-3,0.95) | (0.1,0.1,1000,1) |
| $Al_2O_3$ | (40,80) | 4 | (160,160,160) | (5.8,6.0) | (2e+6,1e+4,1,5e-3,0.95) | (0.1,0.1,1000,1) |
| $C_5H_5N$ | (50,100,100) | 8 | (240,240,240,240) | (6.3,6.5) | (4e+6,2e+4,1,5e-3,0.95) | (0.2,1.0,1000,1) |
| $TiO_2$ | (60,120,120) | 6 | (240,240,240,240) | (7.2,7.5) | (2e+6,1e+4,1,1e-3,0.96) | (0.1,0.1,1000,1) |
| $MoS_2$+Pt | (60,120,120) | 8 | (240,240,240,240) | (7.2,7.5) | (2e+6,1e+4,1,1e-3,0.96) | (1,3,1000,1) |
| HEA | (60,120,120) | 8 | (300,300,300,300) | (6.8,7.0) | (2.3e+6,1e+4,1,1e-3,0.96) | (0.01,10,1000,1) |

[a]The architecture of the embedding/fitting net is given by the number of nodes in each hidden layer. We adopt a ResNet-like structure. For one layer $o^l$ going to the next layer $o^{l+1}$, $o^l$ first undergoes a linear transformation and then a nonlinear activation, i.e., $\tilde{o}^{l+1} = \tanh(W^l o^l + b^l)$, where $\tanh$ is a component-wize hyperbolic tangent function. Next, if the number of nodes in $o^l$ is the same as $o^{l+1}$, let $\hat{o}^{l+1} = o^l$; else, if the number of nodes in $o^l$ is half of $o^{l+1}$, we make two copies of $o^l$s and concatenate them together to define $\hat{o}^{l+1}$. Finally, $o^{l+1} = \hat{o}^{l+1} + t_i \tilde{o}^{l+1}$. In the embedding net we set $t_i$ to 1; while in the fitting net, we set it to a trainable parameter.

[b]$M_2$ the number of columns that one slices from $\mathcal{G}^i$ to define $\mathcal{G}^{i2}$ in Eq. (12).

[c]In the format of $(r_{cs}, r_c)$ defined in Eq. (10).

[d]The parameters $(a_1, a_2, a_3, a_4, a_5)$ in the learning rate (LR) scheme are the total training batch, the decay batch, the batch size, the starting learning rate, and the decay rate, respectively. If the current training batch is $x$, then the learning rate will be $a_4 \times a_5^{x/a_2}$.

[e]The parameters $(p_1, p_2, p_3, p_4)$ in the prefactor (pref.) scheme are the $p_{start}$ for energy, $p_{limit}$ for energy, $p_{start}$ for force, and $p_{limit}$ for force, respectively. They are used to define the loss function in Eq. (13). The prefactor scheme is $p = p_{limit}(1 - a_5^{x/a_2}) + p_{start}(a_5^{x/a_2})$. See definition for $x$, $a_2$ and $a_5$ in the footnote for LR scheme. No virial information is used for all the systems but $C_5H_5N$. For $C_5H_5N$, the $p_{start}$ for virial is 0.2, and $p_{limit}$ for virial is 1.

[f]Mol. uni. means the unified DeepPot-SE model for 7 different molecules.

[g]Mol. ind. means individule DeepPot-SE models for 7 different molecules. The network structures and training schemes are all the same for the 7 cases, so we report in a line.

**Proposition 1.** *The DeepPot-SE model is invariant under translational, rotational, and permutational operations.*

*Proof.* According to the main text, the DeepPot-SE model can be summarized as

$$E^{\boldsymbol{w}}(\mathcal{R}) = \sum_i E^i, \ E^i = E^i(\mathcal{D}^i(\mathcal{R}^i)).$$

It suffices to show that $\mathcal{D}^i(\mathcal{R}^i)$ is invariant under translational, rotational, and permutational operations.

First, since $\mathcal{R}^i$ is composed of relative coordinates between $i$ and its neighbors, $\mathcal{D}^i$ is invariant under translational operations.

Second, note that length scalar $r_{ji}$ and dot product $\boldsymbol{r}_{ji} \cdot \boldsymbol{r}_{ki}$ are invariant under rotational operations. Recall that $\mathcal{D}^i = (\mathcal{G}^{i1})^T \tilde{\mathcal{R}}^i (\tilde{\mathcal{R}}^i)^T \mathcal{G}^{i2}$, where $\mathcal{G}^{i1}$ and $\mathcal{G}^{i2}$ are functions of scalars $r_{ji}$ only and the $jk$-th element of the matrix $\tilde{\mathcal{R}}^i (\tilde{\mathcal{R}}^i)^T$ is $s(r_{ji})s(r_{ki})(1 + \frac{\boldsymbol{r}_{ji} \cdot \boldsymbol{r}_{ki}}{r_{ji} r_{ki}})$, which again involves only scalars and dot products of vectors, we know $\mathcal{D}^i$ is invariant under rotational operations.

Finally, since $[(\tilde{\mathcal{R}}^i)^T \mathcal{G}^{i2}]_{lm} = \sum_k [(\tilde{\mathcal{R}}^i)^T]_{lk} [\mathcal{G}^{i2}]_{km}$, a permutation of atomic indices amounts to a permutation of $k$ in the summation rule $\sum_k$, which leaves $[(\tilde{\mathcal{R}}^i)^T \mathcal{G}^{i2}]_{lm}$ unchanged. Therefore, $(\tilde{\mathcal{R}}^i)^T \mathcal{G}^{i2}$ is invariant under permutational operations. Similarly, $(\mathcal{G}^{i1})^T \tilde{\mathcal{R}}^i$ is also invariant under permutational operations. Therefore, $\mathcal{D}^i$ is invariant under permutational operations. $\square$