[Reviews · NeurIPS 2018]

Reviewer 1



The paper proposes a new learnable model DeepPot-SE for inter-atomic potential energy surfaces (PES) based on deep neural networks. The authors start by introducing a number of requirements that a PES model should fulfil. Compared to other proposed model, this proposed model is the first to fulfil all these requirements, including differentiability and preserving natural symmetries. In the empirical evaluation, the performance of the proposed model is comparable to or better than state-of-the-art models as measured in MAE for energy and force predictions. Generally the paper the paper is well written and easy to follow. However, the paper may be hard to follow for someone with little experience in PES modelling, e.g., it is never explained in the paper what is meant by "Finite and Extended Systems" (which is part of the title of the paper). One of the central aspects of the paper is the feature matrix D^i in equation (11) that is invariant to translations, rotations and permutations. The authors have shown that some of the subexpressions of (11) are indeed invariant. However, given that this feature matrix plays such a central role in the paper, I think that the authors should give some details beyond "It is straightforward to see that D^i, and hence the DeepPot-SE model, preserve all the necessary symmetries" (line 141-142), e.g. in supplementary material. The work seems to be an elegant improvement of the Deep Potential model [17], which unfortunately also makes the work seems a little incremental. However, the idea of the feature matrix D seems novel and could have potential applications outside the domain of PES modelling (as also noted by the authors). Such applications are however not considered. The authors evaluate the model empirically on a wide selection of systems. For small organic molecules, the proposed DeepPot-SE model is compared to three state-of-the-art models. However, given that DeepPot-SE is an improvement of the Deep Potential model, I find it surprising that the Deep Potential model is not included as a baseline. Generally, the DeepPot-SE model has a performance that is comparable to the other models. However, it does not outperform the other models. The results for bulk systems are compared to the DeePMD model, and on these datasets, the proposed consistently outperforms the baseline model. However, for this dataset I am slightly puzzled about the training and test split: 90% of the snapshots of an MD trajectory are randomly selected as training data and 10% are selected as test data. As snapshots are only 10fs apart, I worry that the examples in the training and test set are highly similar. Could the authors please clarify this in their rebuttal. To summarise, the paper is of high quality and clarity. The work has some both incremental aspects and some novel/original aspect (that may have application beyond PES). The results on small organic molecules are not highly significant, however, on bulk systems, the baseline model was systematically outperformed. Minor comments: 1. Figure 1: ".= (b3)" -> ". (b3)" 2. Equation (8): it would be useful to remind the reader that r_ji = $|\vec{r}ji}|$. 3. Figure 2: "DeepMD-SE" -> "DeepPot-SE" Comments after rebuttal: The authors have addressed most of my issue in their rebuttal, and I recommend accepting the paper. However, I would like to stress the importance of added a rigorous proof for that D^i (equation 11) preserves all the intended symmetries in the camera-ready version.

Reviewer 2



A neural network model for the estimation of potential energy surfaces of chemical systems given their atomic configuration is proposed. It is an extension of DeepPot and introduces smoothness to the functional, enabling better training. It incorporates known symmetries (index permutation, rotation, translation) of the systems by building them into the architecture. It is shown that the model can fit a wide variety of chemical systems ranging from small molecules to bulk matter, all within chemical accuracy with respect to DFT. With the extensive amount of experiments and well-described architecture and training, this is a very good contribution.

Reviewer 3



The manuscript proposes a new potential energy surface model designed to be scalable, accurate, and to preserve natural symmetries. The model is tested on a range of systems, and compares favourably with existing methods. Quality: The manuscript is of high technical quality. It contains a good motivation (stating desirable requirements), and then lays out the necessary theory step by step, illustrating the behavior of the model through examples on one system (Figure 1). This is backed up by experiments on a range of different systems, further establishing the competitiveness of the method. Clarity: The paper is well written and well structured. Although I am not an expert in the PES-estimation field I found it fairly easy to follow, and enjoyable to read. Originality: The authors present a convincing case for the originality of their approach, with considerable effort summing up existing efforts in the field. However, I am not comfortable enough within the literature in the PES-estimation field to reliable assess the degree of originality of the approach. Significance: Finding better ways to estimate potential energy surfaces is an important task in molecular modelling, and the ability to efficiently encode relevant symmetries will be an important ingredient for further progress. In my view this manuscript therefore constitutes an important contribution Specific comments: It was not entirely clear to me how requirement 3) (introduced in line 27) is ensured with your model. It seems to me that you consider pairwise distances between atoms (e.g. the double indices in eq 8), which would imply that the method scales quadratically rather then linearly in the number of atoms (which is also what you would expect for a molecular forcefield). Perhaps I misunderstood something. Could you clarify what you mean with linear scaling? Minor: Figure 1, caption: There is a spurious '=' preceding '(b3)' Update after rebuttal period: I only raised minor issues in my review, which the authors have addressed in their rebuttal. I therefore stick with my original (high) rating of the manuscript.